# Investigating the profile of patients with idiopathic inflammatory myopathies in the post-COVID-19 period

Lu Cheng,[1] Yan-hong Li,[2] Yin-lan Wu,[2] Yu-bin Luo,[2] Yu Zhou,[3] Tong Ye,[4] Xiu-ping Liang,[2] Tong Wu,[2] De-ying Huang,[2] Jing Zhao,[2] Yi Liu,[2] Zong-an Liang,[1] Chun-yu Tan[2]

**ABSTRACT** The 2019 coronavirus disease (COVID-19) pandemic has changed the characteristics of many diseases. It remains unclear whether idiopathic inflammatory myopathies (IIMs) exhibit distinct phenotypes in the context of COVID-19. This retrospective study analyzed patients with IIMs admitted to West China Hospital from January 2022 to December 2023. Among them, 171 had a history of COVID-19 (prior COVID-19 [PC]), while 121 did not (no prior COVID-19 [NPC]). Medical histories, lab tests, and echocardiography data were compared. The PC group exhibited a greater incidence of cardiac damage, including a greater proportion of cardiac injury ($P = 0.016$), clinical diagnosis of myocarditis ($P = 0.02$), palpitation ($P = 0.031$), and Myositis Activity Assessment Visual Analog Scale/Myositis Intention-to-Treat Activity Index cardiovascular involvement scores (all $P < 0.001$), and elevated levels of myoglobin ($P = 0.03$), creatinine kinase MB ($P = 0.015$), cardiac troponin T ($P = 0.011$), N-terminal pro-B-type natriuretic peptide ($P = 0.028$), lactate dehydrogenase ($P = 0.033$), and hydroxybutyrate dehydrogenase ($P = 0.019$). Echocardiographic analysis revealed a greater diameter of the left atrium ($P = 0.040$), left ventricle ($P = 0.013$), greater end-diastolic dimension ($P = 0.042$), and greater end-diastolic volume ($P = 0.036$) in the PC group than in the NPC group. Transcriptional data analysis based on public databases indicated that various mechanisms, including collagen matrix proliferation, calcium ion pathway regulation, oxidative stress, cell proliferation, and inflammatory molecules, collectively contribute to the pathogenesis of myocardial damage in patients with IIMs and COVID-19. The study serves as a crucial reminder for clinicians to remain vigilant regarding the enduring cardiovascular consequences associated with IIMs after COVID-19.

**IMPORTANCE** This study systematically analyzed the clinical features, laboratory test results, and echocardiographic findings of patients with IIMs, comparing those with and without a history of COVID-19 infection. The analysis revealed significant alterations in the clinical manifestations of IIM patients after COVID-19, particularly in relation to cardiac involvement. Our findings highlight the crucial importance for clinicians to maintain vigilance concerning the potential long-term cardiovascular sequelae associated with IIMs in post-COVID-19 patients.

**KEYWORDS** post-COVID-19, COVID-19, SARS-CoV-2, idiopathic inflammatory myopathy (IIM), myocardial damage

With the implementation of effective control measures for the 2019 coronavirus disease (COVID-19), the impact of severe acute respiratory syndrome coronavirus type 2 (SARS-CoV-2) is gradually diminishing. However, a large body of research shows that at least 10% of people with acute SARS-CoV-2 infection continue to experience long-term symptoms after initial recovery, and these conditions are referred to as long COVID or post-acute sequelae of COVID-19 (1). Long COVID can involve multiple organ

**Peer Reviewer** Safdar Ali, Cholistan University of Veterinary And Animal Sciences, Bahawalpur, Punjab, Pakistan

Address correspondence to Chun-yu Tan, annaquintessence@163.com, or Zong-an Liang, liangza@scu.edu.cn.

Lu Cheng and Yan-hong Li contributed equally to this article. Authors ranked by mutual agreement.

The authors declare no conflict of interest.

See the funding table on p. 13.

systems, including respiratory system dysfunction, cardiovascular complications, central nervous system dysfunction, and blood clotting dysfunction (1).

Autoimmunity is one of the most important hypothesized mechanisms of long COVID. Various autoantibodies, including anti-nuclear antibodies (ANAs), lupus anticoagulants, anti-2-glycoprotein 1β, and anti-cardiolipin antibodies, have been identified in COVID-19 patients (2). A plethora of reports on autoimmune thrombocytopenic purpura, autoimmune hemolytic anemia, inflammatory arthritis, and vasculitis following COVID-19 further substantiates the potential association between COVID-19 and autoimmune diseases (2, 3).

Idiopathic inflammatory myopathies (IIMs) are a group of systemic autoimmune inflammatory diseases that mainly involve the proximal muscles of the extremities (4). Various viral infections, such as coxsackie B, parvovirus, enterovirus, human T-cell-lymphotropic virus, and human immunodeficiency virus, have been recognized as significant environmental factors in the development of IIMs (2). Following SARS-CoV-2 infection, patients frequently experience symptoms such as myalgia, myasthenia, and elevated creatine kinase levels, and even new-onset IIMs have been reported (3). Although two studies have reported some characteristics of IIMs during the COVID-19 pandemic (5, 6), the small sample sizes of these studies make it difficult to determine whether COVID-19 has altered the clinical manifestations of IIMs. Our study involved 292 patients with IIMs, aimed at comprehensively summarizing the clinical features, laboratory findings, and imaging data, and provided valuable insights into the characteristics of IIM patients in the post-COVID-19 period.

## MATERIALS AND METHODS

### Research design and subjects

In this retrospective study, we recruited all IIM patients who were admitted to West China Hospital between January 2022 and December 2023. All participants were aged 18 years or older and met the 2017 European League Against Rheumatism/American College of Rheumatology criteria for the classification of IIMs (7). Moreover, patients with other connective tissue diseases, malignancies, inherited myogenic disorders, tuberculosis, septicopyemia, the acute phase of COVID-19, and those with missing data on myocardial markers were excluded. Finally, 292 IIM patients who met the inclusion criteria were included in the study cohort. Patients were divided into two groups according to whether they had a history of COVID-19: no prior COVID-19 (NPC, $n = 121$) and prior COVID-19 (PC, $n = 171$). Prior COVID-19 was defined as patients who had symptoms of viral infection, such as fever, myalgia, sore throat, and cough and a positive SARS-CoV-2 nucleic acid, and the SARS-CoV-2 nucleic acid turned negative for at least 4 weeks after treatment or no treatment. The evaluation and inclusion process for these patients is illustrated in Fig. 1.

### Collection of clinical features

All the data were collected from the electronic medical records database of West China Hospital. The medical data collected included demographic characteristics, such as age, gender, duration of IIMs, duration of COVID-19, clinical symptoms and signs, including fever, cough, expectoration, dyspnea, arthritis, myalgia, myasthenia, and rash, and laboratory features, including routine blood parameters, erythrocyte sedimentation rate, C-reactive protein level, alanine aminotransferase level, aspartate aminotransferase level, alkaline phosphatase level, gamma glutamyl transferase (GGT), creatine kinase level, lactate dehydrogenase (LDH) level, hydroxybutyrate dehydrogenase (HBDH) level, myoglobin (MYO) protein level, cardiac troponin T (cTnT) level, N-terminal pro-B-type natriuretic peptide (NT-proBNP) level, creatine kinase isoenzyme MB (CK-MB) level, triglyceride level, cholesterol level, fibrinogen level, anti-thrombin III activity, ANAs, anti-Ro52 antibody, myositis specific antibodies, and echocardiographic parameters.

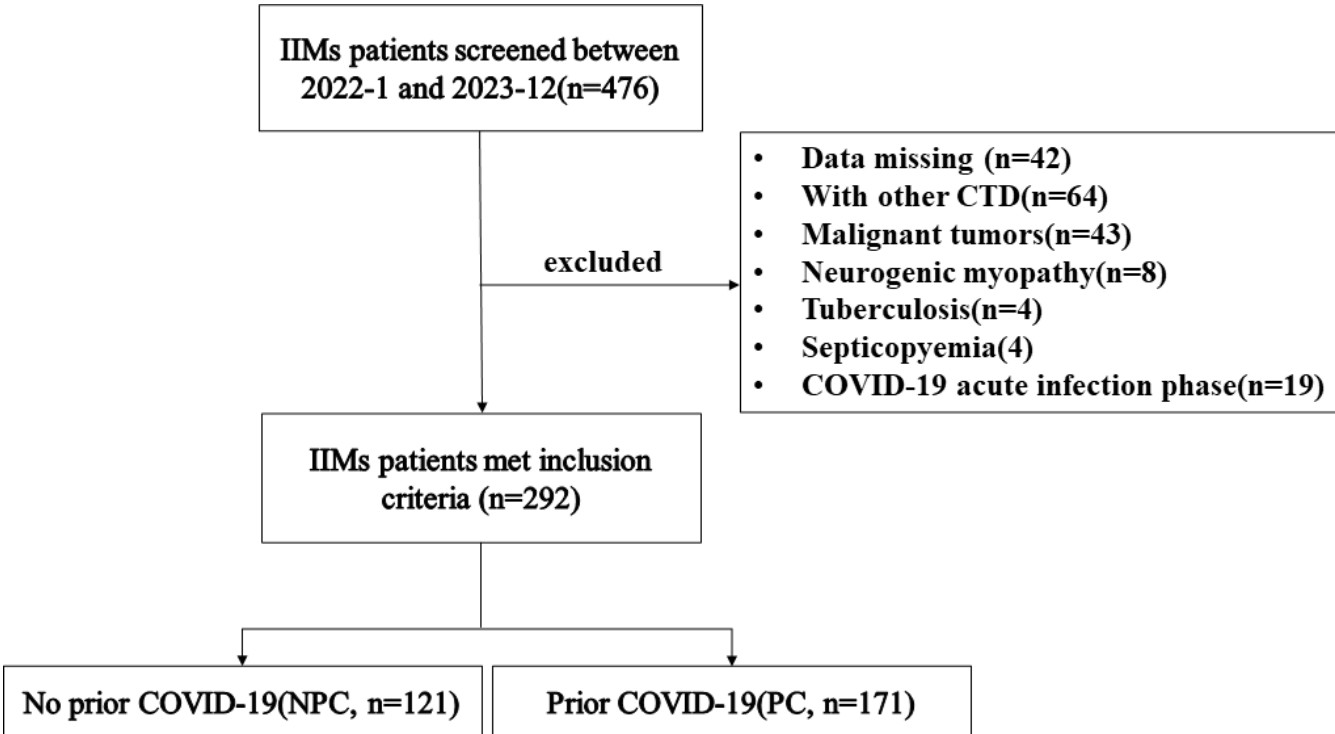

**FIG 1** Flowchart of the clinical study participants. COVID-19, coronavirus disease 2019; CTD, connective tissue disease; IIM, idiopathic inflammatory myopathy.

## Myositis disease activity assessment

Disease activity scores were performed by two experienced rheumatologists based on medical records. Disease activity for IIMs was measured using the Myositis Activity Assessment Visual Analog Scale (MYOACT) and the Myositis Intention-to-Treat Activity Index (MITAX), which were established by the International Myositis Assessment and Clinical Studies group. Detailed scoring methods can be found in our previous article (8).

## Definition of cardiac injury

Cardiac injury is diagnosed according to established criteria derived from prior research, encompassing at least one of the following parameters: (i) elevation of cardiac bio-markers exceeding the 99th percentile upper reference limit; (ii) newly detected electrocardiographic abnormalities, including but not limited to arrhythmias (atrial/ventricular tachycardias and atrial/ventricular fibrillations), conduction disturbances, ischemic changes, repolarization abnormalities, or QT interval prolongation; and (iii) recent echocardiographic findings demonstrating left ventricular systolic dysfunction, impaired ventricular wall motion, pericardial fluid accumulation, or elevated pulmonary arterial pressure (9).

## Analysis of differentially expressed genes

We retrieved the COVID-19 patient data set (GEO accession ID: GSE151879), which used the high-throughput sequencing Illumina NextSeq 500 platform to detect RNA sequences, from which we sorted the myocardial tissue samples from three COVID-19 patients and three normal people (10). The dermatomyositis (DM) patient data set (GEO accession ID: GSE143323), including muscle samples from 39 DM patients and 20 healthy controls, was extracted from RNA sequences using the high-throughput sequencing Illumina HiSeq 3000 platform (11). Using R software (v.2) along with the "DESeq2" and "edgeR" packages, we analyzed the COVID-19 data set (GSE151879) and DM data set (GSE143323). A threshold of |log2 fold change| of >0.585 and adjusted

P value of <0.05 was used as the implementation standard to find the differentially expressed genes (DEGs) of the two data sets. The "Venn" package in R software (v.4.2.1) was used to identify the common DEGs that were differentially expressed and upregulated or downregulated in the two data sets at the same time. Gene Ontology (GO) enrichment analysis and Kyoto Encyclopedia of Genes and Genomes analysis were performed on DEGs using the enrichment database to reveal the functions of DEGs. The "cluster Profiler" package in R software (v.4.2.1) was used to screen the common DEGs of potential biological functions and physiological pathways. GO terms included three parts: biological process, cellular component, and molecular function. A P value of <0.05 and a q value of <0.25 were used as standardized indices.

## Statistical analysis

SigmaPlot (v.12.5) and/or GraphPad Prism (v.8.0.2) software were used for data analysis. Continuous variables are presented as the mean ± standard deviation or median (Q25, Q75), depending on whether they fit a normal distribution. Categorical variables are described statistically using percentages. When the normality (Shapiro–Wilk) test was passed, an independent-sample t-test and one-way analysis of variance were conducted for two-group or multiple comparisons, respectively; otherwise, the Mann–Whitney U test was used for comparison. Chi-square tests were used to compare categorical variables. All tests were bilateral, and P < 0.05 was considered to indicate statistical significance.

## RESULTS

### Baseline demographics

The demographics and clinical characteristics of patients are presented in Table 1. As observed, the general baseline characteristics, including age, sex, body mass index, and duration of IIMs, were comparable between the two groups (Table 1). The proportions of patients with newly diagnosed IIMs were 84 (69.42%) and 109 (63.74%) in the NPC and PC groups, respectively, showing no significant difference (P > 0.05). In the PC group, the time elapsed from prior SARS-CoV-2 infection ranged from 4 weeks to 11 months (data not shown), with an average duration of 5.92 ± 3.18 months. The proportion of patients with myocarditis complications in the PC group was significantly greater than that in the NPC group (12.87% vs. 4.13%, P = 0.020). The median time from COVID-19 to the diagnosis of myocarditis was 6 (3, 9) months (data not shown in the paper). The percentage of cardiac injury in the PC group was significantly higher than that in the NPC group (77.19% vs. 63.64%, P = 0.016). There were no statistically significant differences in complications of interstitial lung disease, comorbidity incidence, and treatment between the two groups (P > 0.050).

### Effects of COVID-19 on the clinical characteristics of patients with IIMs

We analyzed the clinical symptoms of patients with IIMs and observed that the PC group exhibited a greater prevalence of cardiopulmonary symptoms than did the NPC group (Table 1). These symptoms included palpitation (21.05% vs. 10.74%, P = 0.031) and respiratory rate (20.56 ± 2.10 vs. 20.04 ± 0.98, P = 0.048). The NPC group had more rashes (69.42% vs. 52.05%, P = 0.004) and arthritis/arthralgia (40.50% vs. 26.9%, P = 0.021) than did the PC group.

Furthermore, disease activity was assessed in both groups (Fig. 2). Compared to those in the NPC group, patients in the PC group had greater MYOACT/MITAX cardiovascular involvement scores (6 [0, 7] vs. 0 [0, 6], P < 0.001; 9 [0, 9] vs. 0 [0, 9], P < 0.001), global MYOACT/MITAX scores (0.37 ± 0.12 vs. 0.35 ± 0.11, P = 0.046; 0.29 [0.21, 0.40] vs. 0.27 [0.18, 0.35], P = 0.043), and MYOACT pulmonary involvement scores (4.97 ± 3.37 vs. 4.27 ± 3.35, P = 0.033). The MYOACT/MITAX cutaneous involvement scores in the NPC group were greater than those in the PC group (6 [0, 7] vs. 5 [0, 7], P = 0.035; 3 [0, 3] vs. 1 [0, 3],

**TABLE 1** Baseline characteristics of IIM patients

| Characteristics | NPC (n = 121) | PC (n = 171) | P |
|---|---|---|---|
| Female sex, n (%) | 95 (79.24) | 126 (73.68) | 0.329 |
| BMI (kg/m$^2$)$^a$ | 22.22 (19.83, 24.59) | 22.48 (20.44, 25.00) | 0.331 |
| Age (years) | 51.00 (45.40, 57.05) | 52.00 (44.00, 58.00) | 0.472 |
| First diagnosed IIMs, n (%) | 84 (69.42) | 109 (63.74) | 0.377 |
| Disease duration (months) | 6.00 (3.00, 24.00) | 10.00 (3.00, 24.00) | 0.489 |
| Time since prior COVID-19 (months)$^b$ | – | 5.92 ± 3.18 | – |
| Comorbidities, n (%) | | | |
| Hypertension | 11 (9.09) | 20 (11.70) | 0.604 |
| Diabetes | 16 (13.22) | 21 (12.28) | 0.952 |
| Coronary artery disease | 5 (4.13) | 8 (4.68) | 0.948 |
| Hyperlipidemia | 33 (27.27) | 37 (21.64) | 0.331 |
| Cerebral infarction | 1 (0.83) | 0 | – |
| COPD | 2 (1.65) | 2 (1.17) | 0.872 |
| Interstitial lung disease | 70 (57.85) | 101 (59.06) | 0.931 |
| Chronic kidney disease | 1 (0.83) | 2 (1.17) | 0.762 |
| Chronic liver disease | 3 (2.48) | 9 (5.26) | 0.378 |
| Anemia | 24 (19.84) | 23 (13.45) | 0.193 |
| Clinical manifestation, n/total (%) | | | |
| Fever | 23 (19.01) | 37 (21.64) | 0.689 |
| Loss of weight | 33 (27.27) | 60 (35.09) | 0.199 |
| Fatigue | 91 (75.21) | 131 (76.61) | 0.891 |
| Chest pain | 6 (4.96) | 16 (9.36) | 0.239 |
| Palpitation | 13 (10.74) | 36 (21.05) | 0.031$^c$ |
| Shortness of breath/dyspnea | 70 (57.85) | 115 (67.25) | 0.129 |
| Rash | 84 (69.42) | 89 (52.05) | 0.004$^c$ |
| Weakness in the proximal muscles | 59 (48.76) | 80 (46.78) | 0.830 |
| Myodynia | 43 (35.54) | 68 (39.77) | 0.541 |
| Arthritis/arthralgia | 49 (40.50) | 46 (26.90) | 0.021$^c$ |
| Dysphagia | 30 (24.79) | 42 (24.56) | 0.926 |
| Cough | 59 (48.76) | 77 (45.03) | 0.610 |
| Expectoration | 53 (43.80) | 69 (40.35) | 0.639 |
| Heart rate (beats/minute) | 91.63 ± 14.31 | 89.42 ± 15.11 | 0.209 |
| Respiratory rate (breath/minute) | 20.04 ± 0.98 | 20.56 ± 2.10 | 0.048$^c$ |
| Clinical diagnosis myocarditis, n (%) | 5 (4.13) | 22 (12.87) | 0.020$^c$ |
| Cardiac injury, n (%) | 77 (63.64) | 132 (77.19) | 0.016$^c$ |
| Treatments | | | |
| Glucocorticoids | 50.00 (35.00, 50.00) | 45.00 (30.00, 50.00) | 0.400 |
| Cyclophosphamide | 27 (22.31) | 48 (28.07) | 0.331 |
| Methotrexate | 11 (9.09) | 18 (10.53) | 0.837 |
| Mycophenolate mofetil | 10 (8.26) | 14 (8.70) | 0.931 |
| Ciclosporin A/tacrolimus | 42 (34.71) | 59 (34.50) | 0.930 |
| Tofacitinib/baricitinib | 12 (9.92) | 20 (11.70) | 0.772 |
| Rituximab | 0 | 3 (1.75) | – |
| Gamma globulin | 11 (9.09) | 22 (12.87) | 0.415 |
| Plasmapheresis | 0 | 2 (1.17) | – |
| ACEIs/ARBs | 4 (3.31) | 7 (4.09) | 0.971 |

$^a$The body mass index (BMI) is the weight in kilogram divided by the square of the height in meter.
$^b$The time from improvement of COVID-19-related symptoms and negative throat swab results to admission.
Abbreviations: ACEI, angiotensin-converting enzyme inhibitor; ARB, angiotensin receptor inhibitor; COPD, chronic obstructive pulmonary disease; NPC, no prior COVID-19; PC, prior COVID-19.
$^c$Indicates a significant difference between two groups; *$P < 0.05$, **$P < 0.01$.

$P = 0.027$). There was no statistical difference in the MYOACT and MITAX scores of other organs (Fig. S1).

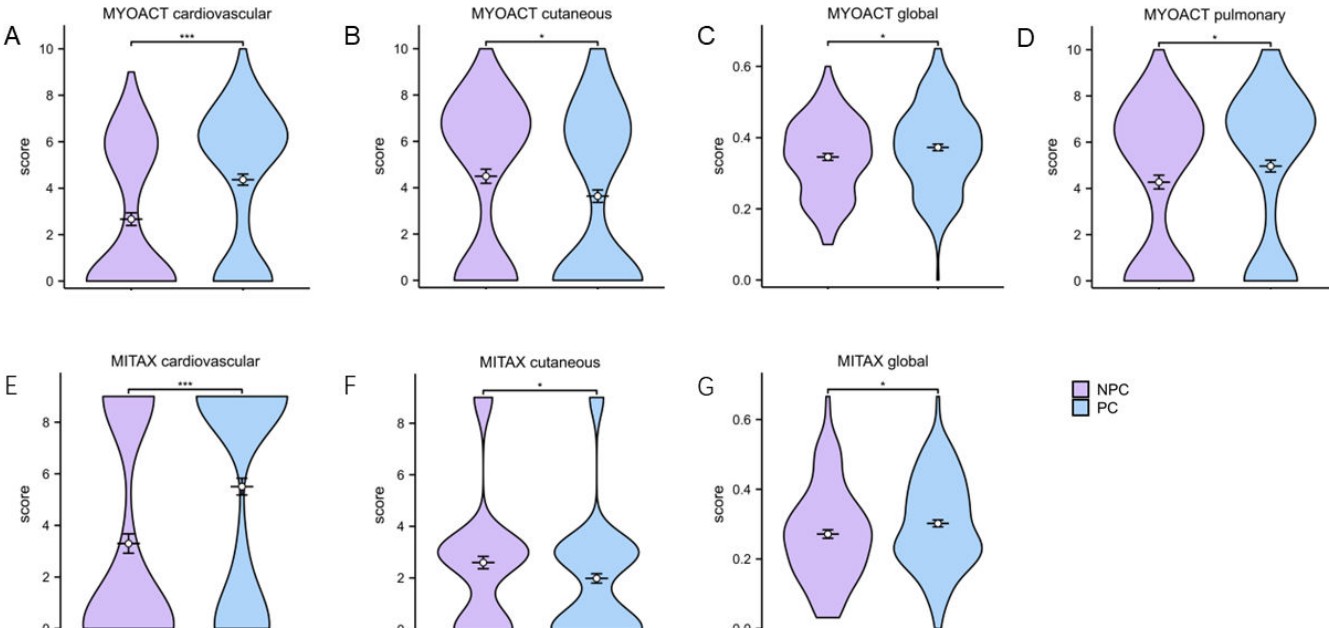

**FIG 2** Myositis disease activity score in IIM patients. MITAX, Myositis Intention-to-Treat Activity Index; MYOACT, Myositis Activity Assessment Visual Analog Scale. Asterisks indicate statistical difference between two groups. *$P < 0.05$, ***$P < 0.001$.

Subsequently, we analyzed the laboratory characteristics of the two groups, as presented in Fig. 3; Fig. S2. The ratio of anti-aminoacyl transfer RNA synthetase antibody (ARS) in the PC group was greater than those in the NPC group (24.56% vs. 11.57%, $P = 0.009$; Fig. 3A). There were no significant differences in the rates of positivity for anti-melanoma differentiation-associated gene 5 antibody (MDA5), anti-nuclear matrix protein 2 antibody, anti-helicase protein antibody, anti-small ubiquitin-like modifier activating enzyme, anti-transcription intermediary factor 1γ, anti-3-hydroxy-3-methyglutaryl coenzyme A reductase, anti-signal recognition particle, or ANA between the two groups (Fig. 3A and B). The percentage of patients who were positive for anti-Ro52 antibody in the PC group was greater than that in the NPC group (49.69% vs. 36.98%, $P = 0.046$; Fig. 3C). Significantly, markers of myocardial injury, such as MYO (128.50 [26.70, 620.25] vs. 53.25 [21.82, 462.50]; $P = 0.030$), CK-MB (7.32 [1.64, 70.50] vs. 3.04 [1.18, 21.52]; $P = 0.015$), cTnT (54.50 [15.90, 153.00] vs. 22.00 [9.65, 101.35]; $P = 0.011$), NT-proBNP (133.00 [61.50, 322.50] vs. 97.50 [41.75, 232.50]; $P = 0.028$), LDH (360.00 [250.00, 590.00] vs. 316.00 [234.50, 488.50]; $P = 0.033$), and HBDH (274.00 [194.00, 429.00] vs. 231.00 [176.00, 366.00]; $P = 0.019$), were significantly greater in the PC group than those in the NPC group (Fig. 3D through I). Moreover, the GGT levels were significantly greater in the PC group (41.00 [19.00, 98.00] vs. 28.00 [18.00, 60.50]; $P = 0.048$; Fig. 3J). In addition, the white blood cell and neutrophil counts in the PC group were significantly greater than those in the NPC group (7.44 [5.48, 9.75] vs. 6.40 [4.46, 9.11], $P = 0.018$; 5.30 [3.76, 7.18] vs. 4.39 [2.89, 7.27], $P = 0.023$; Fig. 3K and L). Interestingly, the fibrinogen levels (2.67 [2.21, 3.38] vs. 3.03 [2.40, 4.08]; $P = 0.002$) and platelet counts (198.00 [157.00, 246.00] vs. 219.00 [174.50, 261.50]; $P = 0.014$) in the PC group were significantly lower than those in the PC group (Fig. 3M and N). There was no significant difference between the two groups in other laboratory measures (Fig. S2). These findings suggest that patients with IIMs who have previously contracted COVID-19 display altered clinical symptoms and laboratory test results, particularly pertaining to cardiac characteristics.

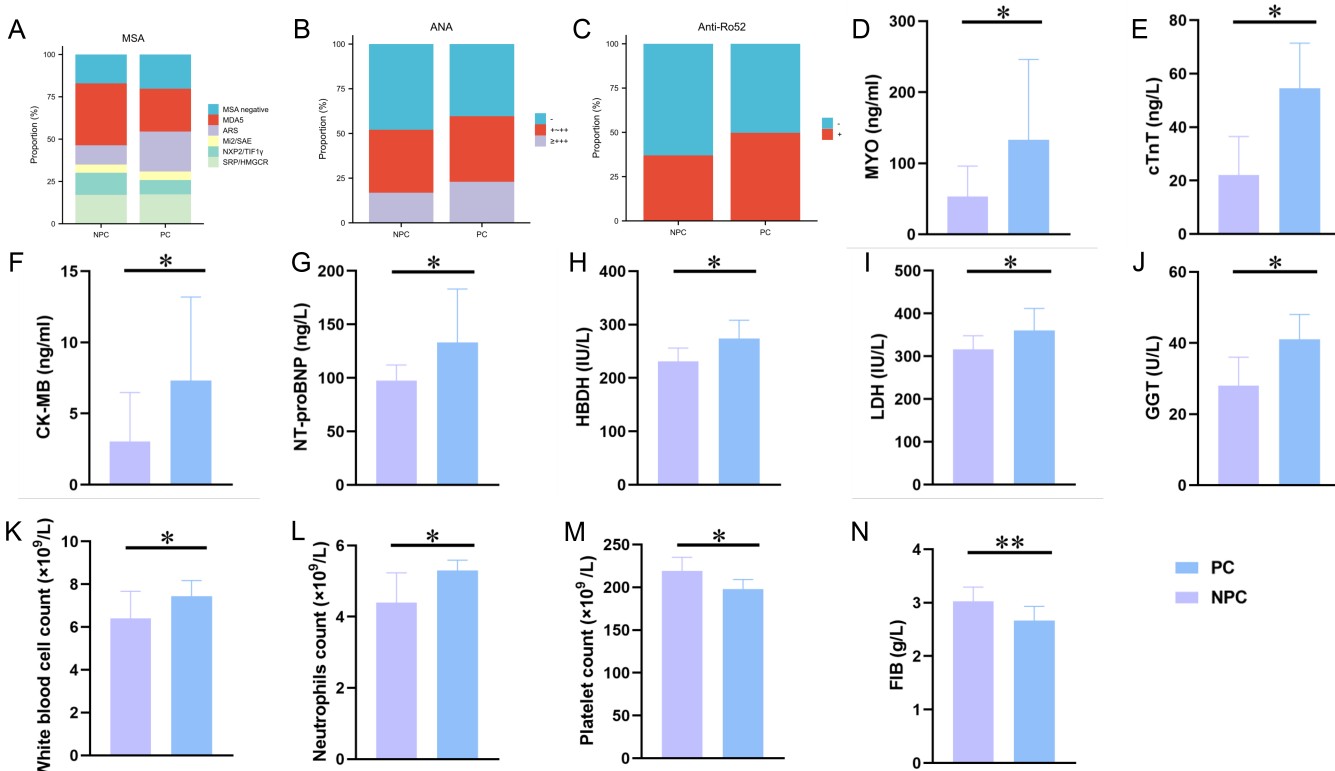

**FIG 3** Laboratory findings of IIM patients on admission to the hospital. (A) Distribution of myositis-specific antibodies (MSA) in IIM patients. ARS, anti-aminoacyl transfer RNA synthetase antibody; HMGCR, anti-3-hydroxy-3-methyglutaryl coenzyme A reductase; MDA5, anti-melanoma differentiation-associated gene 5 antibody; Mi2, anti-helicase protein antibody; NXP2, anti-nuclear matrix protein 2 antibody; SAE, anti-small ubiquitin-like modifier activating enzyme; SRP, anti-signal recognition particle; TIF1γ, anti-transcription intermediary factor 1γ. (B) Anti-nuclear antibody (ANA) expression in IIM patients. −, ANA negative; +~++, 1:100 ≤ ANA ≤ 1:320; +++, ANA ≥ 1:1,000. (C) Anti-cytoplasmic ribonucleoprotein of 52 kDa (Ro52) expression in IIM patients. (D–G) Markers of myocardial injury expression in IIM patients. CK-MB, creatinine kinase MB; cTnT, cardiac troponin T; MYO, myoglobin; NT-proBNP, N-terminal pro-B-type natriuretic peptide. (H–J) Blood biochemical indexes of IIM patients. GGT, gamma glutamyltransferase; HBDH, hydroxybutyrate dehydrogenase; LDH, lactate dehydrogenase. (H–M) Blood cells from patients with IIMs. (N) Coagulation index in IIM patients. FIB, fibrinogen. Asterisks indicate statistical difference between two groups. *$P < 0.05$, **$P < 0.01$.

## Echocardiographic parameters exhibited alterations in IIM patients with a history of COVID-19

To further investigate the impact of COVID-19 on cardiac function in patients with IIMs, we analyzed echocardiography parameters, including measures of cardiac structure and function (Fig. 4; Fig. S3). Due to missing echocardiography data, there were 111 and 149 patients in the NPC and PC groups, respectively. Cardiac structural parameters revealed that the left ventricular diameter (left ventricle) and left atrial diameter (left atrium) were greater in the PC group than in the NPC group (46.43 ± 3.68 vs. 45.52 ± 3.25, $P = 0.040$; 33.00 [30.00, 36.00] vs. 31 [28.00, 34.00], $P = 0.013$; Fig. 4A and B). The end-diastolic volume was higher in the PC group than in the NPC group (99.49 ± 18.73 vs. 94.81 ± 15.76, $P = 0.036$; Fig. 4C), while the end-diastolic dimension was greater in the PC group than in the NPC group (47.00 [44.00, 49.00] vs. 45.00 [44.00, 48.00], $P = 0.042$; Fig. 3D). No significant differences were observed for the other echocardiographic parameters examined (Fig. S3). In addition, we analyzed the electrocardiograph characteristics of patients with IIMs, and there was no statistically significant difference in the rate of arrhythmia between the two groups (Table S1).

To ascertain the potential association between the extent of cardiac damage in individuals with post-COVID-19 IIMs and the duration elapsed since SARS-CoV-2 infection. Patients with IIMs were classified according to the time since previous SARS-CoV-2 infection. Table S2 showed that IIM patients with a short period of acute COVID-19

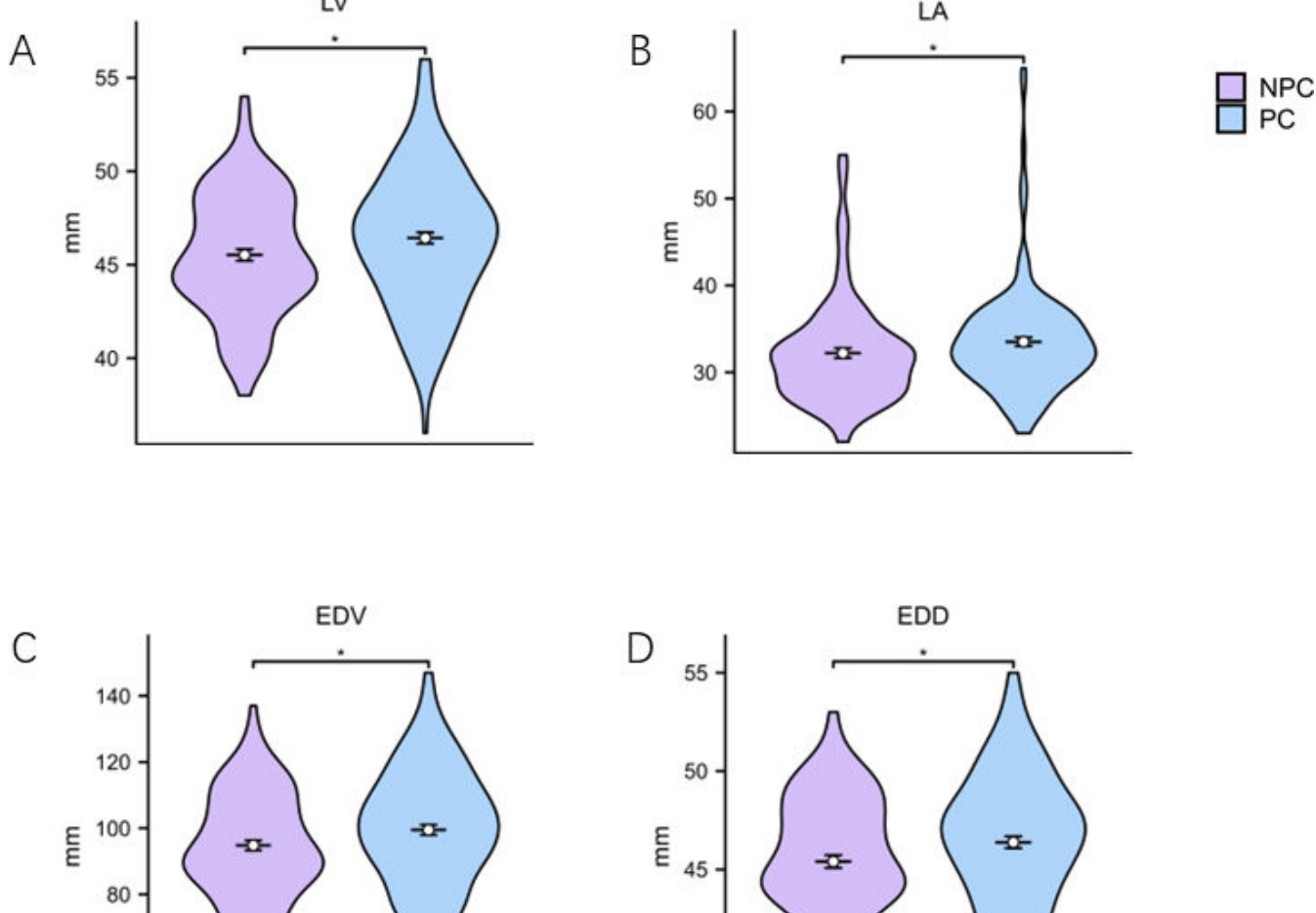

**FIG 4** Echocardiographic characteristics of IIM patients. EDD, end-diastolic dimension; EDV, end-diastolic volume; LA, left atrium; LV, left ventricle. Asterisks indicate a significant difference between two groups. *$P < 0.05$.

had a higher MYOACT/MITAX global score and the highest MYOACT pulmonary score (all $P < 0.05$), while there were no statistically significant differences in cardiac damage-related indicators.

## The putative mechanism underlying COVID-19-induced cardiac damage in patients with IIMs

To investigate the potential mechanism of cardiac injury in patients with IIMs in the post-COVID-19 era, we employed RNA-seq technology coupled with bioinformatics analysis to elucidate disparities in gene expression profiles between DM patients and COVID-19 patients. By examining the number of DEGs between DM patients and COVID-19 patients, we identified 7,189 DEGs in DM patients (Fig. 5A) comprising 5,654 upregulated genes (depicted as red dots) and 1,679 downregulated genes (depicted as blue dots). Conversely, COVID-19 patients exhibited 3,080 DEGs (Fig. 5B), including 1,278 upregulated genes (red dots) and 1,802 downregulated genes (blue dots). A comparison of DEGs between DM patients and COVID-19 patients revealed an overlap of 720 genes (Fig. 5C). To explore the underlying mechanisms and pathways associated with these DEGs within our data sets, functional enrichment analyses using Gene Ontology terms

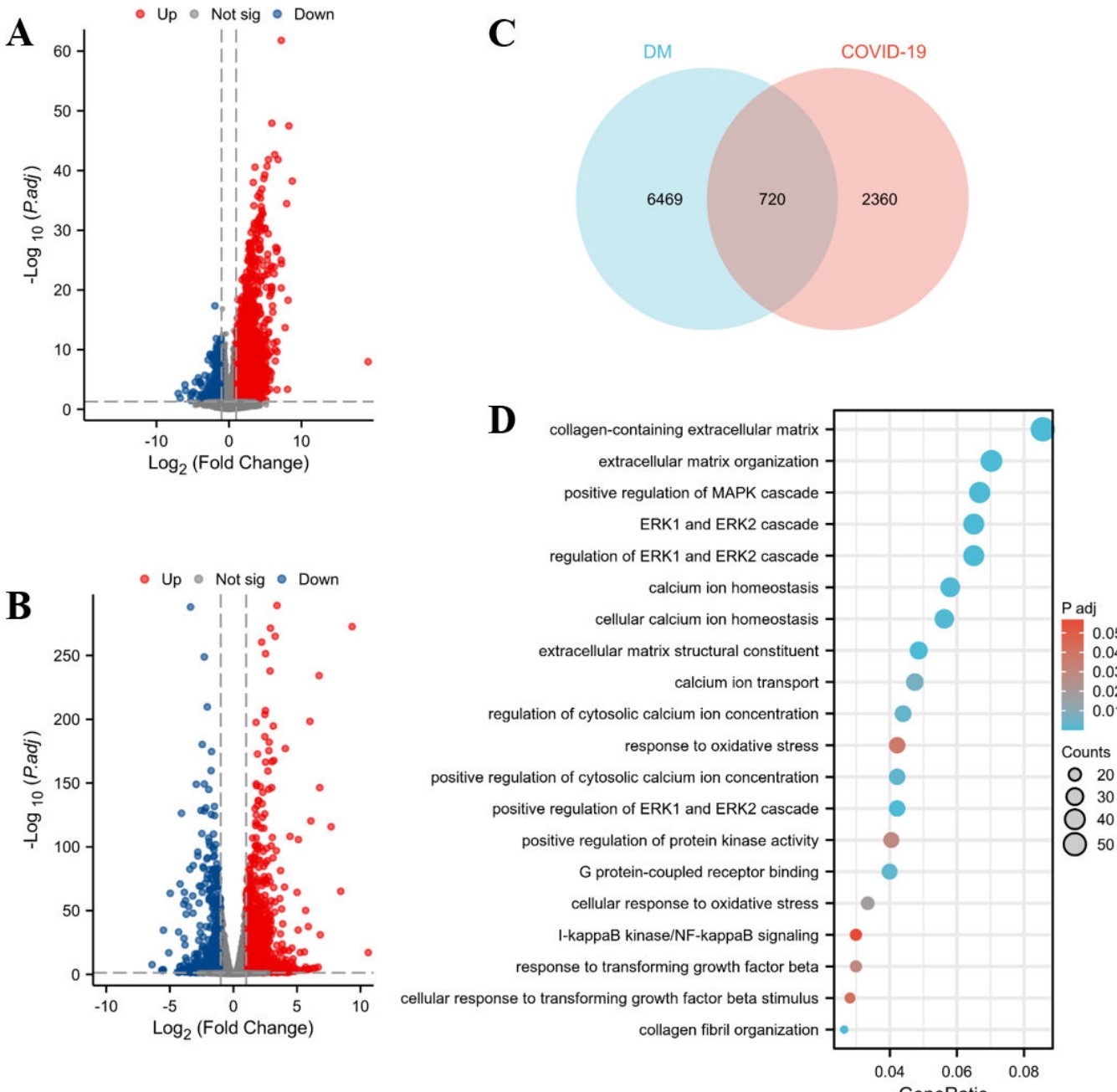

**FIG 5** Bioinformatics analysis of the transcriptomes of muscle tissues from DM patients and myocardial tissues from COVID-19 patients. (A) Differently expressed genes in the DM data set (GSE143323) in the volcanic figure. (B) Differentially expressed genes (DEGs) in the COVID-19 data set (GSE151879) in the volcanic figure. (C) Venn diagram showing the overlap of DEGs between COVID-19 patients and DM patients. (D) Bubble plot of GO and Kyoto Encyclopedia of Genes and Genomes functional enrichment analysis results of common DEGs between COVID-19 patients and DM patients.

and Kyoto Encyclopedia of Genes and Genomes were performed. The results demonstrated that biological processes and enriched pathways related to collagen matrix proliferation, calcium ion pathway regulation, oxidative stress response, cell proliferation, and cell inflammatory molecules were significantly enriched among these targets (Fig. 5D).

## DISCUSSION

The COVID-19 pandemic has changed the patterns and characteristics of many diseases, including diabetes and cardiovascular disease (12–14). A mounting body of research has focused on elucidating the association between COVID-19 and autoimmune diseases (2, 3). Unfortunately, few studies have provided a detailed description of the characteristics of IIM presentation in post-COVID-19 patients. This retrospective study aimed to delineate IIMs in the context of the post-COVID-19 period, with a specific focus on identifying significant alterations in these patients, especially with respect to cardiac damage.

Our study revealed that myocardial damage, including increased incidence of palpitations, increased heart disease activity score, increased myocardial marker levels, and deterioration of cardiac structure as indicated by echocardiography, was the most prominent manifestation of post-COVID-19 infection in IIM patients, as did the number of IIMs with a clinical diagnosis of myocarditis. Cardiac damage is an important complication of IIMs and affects the mortality of IIM patients (15). In fact, the cardiac involvement of IIM patients is relatively insidious, and most patients often have no clinical symptoms, only elevated myocardial enzymes, cardiac magnetic resonance imaging (MRI), and/or myocardial biopsy abnormalities, known as subclinical cardiac involvement (15). The reported proportion of patients with IIMs with cardiac damage ranges from 9% to 72% (16). The cardiovascular effects of the evolution of SARS-CoV-2 infection are among the most prominent and urgent concerns (17). According to a multicentered prospective study of patients with myocardial damage during COVID-19, myocardial damage occurs in one-third of patients with COVID-19 and is associated with an increased risk of death (18). Even myocardial damage can persist until the recovery period (19). The American College of Cardiology has provided detailed explanations and recommendations for adults with cardiovascular sequelae post-COVID-19 (20). However, few studies have focused on cardiac damage in patients with post-COVID-19 IIMs. Previous studies have indicated that the concurrent presence of myositis and inflammatory cardiac disease in COVID-19 patients is characterized by a younger age group and fewer respiratory symptoms (21, 22). In individuals with newly diagnosed anti-synthetase syndrome (ASS) following SARS-CoV-2 infection, clinical manifestations are limited to myocardial and skeletal muscle involvement (23). More interestingly, no significant difference was found in respiratory or skeletal muscle damage between the two groups in our study. These results indicate that SARS-CoV-2 has the potential to alter long-term trajectories associated with cardiac suffering among individuals affected by IIMs.

There are many hypotheses about the mechanism of cardiac damage caused by SARS-CoV-2, including SARS-CoV-2 directly attacking myocardial cells or endothelial cells, SARS-CoV-2 infection triggering cytokine storms caused by immune dysregulation, and hypercoagulable state leading to cardiac damage (17, 18, 20). SARS-CoV-2 enters host cells through binding to angiotensin-converting enzyme 2 (ACE2) on the cell surface to cause direct damage, and cardiomyocytes express high levels of ACE2 on the surface, so they are easily affected. Moreover, myocardial injury after SARS-CoV-2 infection was associated with cytokines in peripheral blood in a large retrospective study; myocarditis was identified in autopsy studies; and microthrombi were identified on cardiac MRI in patients who recovered from COVID-19 (18, 24, 25). Interestingly, our data showed that platelet and fibrinogen levels were lower in patients with IIMs during the post-COVID-19 period, which may suggest that abnormal coagulation is the mechanism of myocardial damage in patients with IIMs. Our findings have been confirmed in other studies of long COVID patients. Martins-Gonçalves et al. investigated platelet function in post-COVID-19 recovery patients and discovered persistent platelet activation and hyperactivity compared to healthy controls (26). An 18-month follow-up study on peripheral blood clotting status among individuals recovering from COVID-19 revealed an enduring procoagulant state associated with persistent symptoms (27). Complement system activation is an important mechanism of coagulation abnormalities (28). Currently, the management of cardiac complications in patients with IIMs remains

poorly standardized, with corticosteroid therapy in combination with immunosuppressive agents serving as the cornerstone treatment for associated myocarditis (15). While the ACE2 receptor has been identified as a crucial pathway in COVID-19-mediated cardiac injury, and preclinical studies have demonstrated that ACE2 modulation therapy can elevate ACE2 expression levels, current clinical evidence does not substantiate any association between angiotensin-converting enzyme inhibitor (ACEI)/angiotensin receptor inhibitor (ARB) use and increased susceptibility to COVID-19 infection, nor with elevated risks of severe disease progression or mortality in COVID-19 patients (29). Continued use of ACEIs/ARBs is recommended for patients with cardiovascular damage from COVID-19 (29).

To further investigate the potential mechanisms of myocardial injury in patients with IIMs during the post-COVID-19 period, we employed transcriptomics and bioinformatics analysis. The enrichment analysis of DEGs revealed that collagen matrix proliferation, calcium ion pathway regulation, oxidative stress, cell proliferation, and inflammatory molecules were key factors contributing to cardiac damage in IIM patients infected with SARS-CoV-2. Our bioinformatics analysis results are consistent with the hypothesized mechanisms underlying COVID-19-associated heart injury. The persistence of the virus in cardiac tissue triggers inflammation leading to myocardial fibrosis; alternatively, viral-induced changes in host proteins can cross-react via molecular mimicry mechanisms (17). A study investigating postmortem examinations of COVID-19 fatalities reported widespread muscle and myocardial damage despite low or negative viral loads in tissues, suggesting a possible role for immune-mediated myocardial injury (30). In conclusion, cardiac injury following SARS-CoV-2 infection is likely multifactorial; therefore, a comprehensive understanding of these mechanisms is crucial for improving diagnosis and treatment processes. Further clinical and basic research efforts are urgently needed to explore this topic.

In addition, our data present several additional features of IIMs in the post-COVID-19 period. The proportion of serum ARS is significantly increased in IIM patients. At present, it is generally believed that the serum antibodies of IIM patients are closely related to clinical subtype, so we analyzed the subtype composition of IIMs (Table S3). We found structural changes in the IIM subtypes after SARS-CoV-2 infection, with a lower proportion of patients with dermatomyositis and anti-MDA5-positive dermatomyositis and a greater proportion of patients with ASS and unclassified IIMs. Case reports have documented the emergence of IIMs following the COVID-19 pandemic (3, 23, 31). Most importantly, COVID-19 and anti-MDA5-positive dermatomyositis have notable similarities in clinical features and pathogenesis, including comparable pulmonary interstitial lesions, activation of the IFN-I signaling pathway, and response to immunosuppressive therapy (32, 33). These findings suggest that SARS-CoV-2 may serve as an environmental risk factor for the onset of IIMs (34). Previous studies have reported that the proportion of COVID-19 patients who are anti-MDA5 antibody positive is increased, and it is a marker of severe disease and poor prognosis (35). However, we found that the proportion of post-COVID-19 patients with anti-MDA5 positive dermatomyositis did not increase. This suggests that not all patients with COVID-19 who are positive for anti-MDA5 antibodies are diagnosed with dermatomyositis and that the mechanisms underlying the two diseases are potentially different. In addition, this result may also be due to survivor bias. The increased incidence and hospitalization rate of anti-synthetase syndrome in post-COVID-19 patients have also been reported in other studies, but the specific mechanism is still unclear (36, 37).

Elevated bile duct enzymes are another feature of patients with IIMs in the post-COVID-19 period. Increased expression of ACE2 in the biliary epithelium increases susceptibility to SARS-CoV-2 infection, leading to chronic liver disease and providing insights into the potential molecular mechanism underlying biliary tract damage caused by COVID-19 (38, 39). Roth et al. characterized liver biopsies from three individuals who experienced cholestasis after recovering from COVID-19 and concluded that post-COVID-19 cholangiopathy represents a distinct form of liver injury (39). Patients

with IIMs in the post-COVID-19 period show increased white blood cell and neutrophil counts, while there was no significant difference in the dosage of glucocorticoids. This phenomenon has been mentioned in other studies of long COVID (40, 41). The specific mechanism is not clear and may be related to the chronic inflammatory state and stress state of patients. However, further prospective studies are needed to explore the pathogenesis and clinical outcomes of IIMs in this context.

Our study also has several limitations. First, this was a retrospective study conducted at a single center, and all the data were obtained from the electronic medical record database. Detailed information regarding the COVID-19 status of these patients was not available, preventing us from determining any potential associations with subsequent cardiac injury. Second, more specific detection methods, such as cardiac MRI and myocardial biopsy, are required to accurately assess cardiac damage in these patients. In addition, the absence of other connective tissue diseases and healthy volunteer controls in this study does not confirm that myocardial damage is specific in patients with IIMs. Last, more than anything, it remains unclear whether the mechanism of disease evolution of IIMs after the COVID-19 pandemic differs fundamentally from that before the outbreak.

## Conclusions

The clinical manifestations of patients with IIMs following COVID-19 exhibit notable alterations, particularly in terms of cardiac impairment. Our findings serve as a crucial reminder for clinicians to remain vigilant regarding the enduring cardiovascular consequences associated with IIMs subsequent to COVID-19.

## ACKNOWLEDGMENTS

We thank AJE Organization for the English language editing.

This work was supported by the Sichuan Science and Technology Program (2020YFS0005) and the International Cooperation Program of Science and Technology Department of Sichuan Province (2023YFH0081).

The authors have no financial relationships relevant to this article.

L.C.: writing (review, editing, and original draft); visualization, and validation; Y.-H.L.: writing (original draft) and formal analysis; Y.-L.W.: methodology, formal analysis, and data curation; Y.-B.L.: supervision and conceptualization; Y.Z.: resources; T.Y.: data curation; X.-P.L., T.W., D.-Y.H., and J.Z.: investigation; Y.L.: project administration; Z.-A.L.: supervision, funding acquisition, and conceptualization; C.-Y.T.: supervision, funding acquisition, and conceptualization.

Written informed consent was obtained from all individual participants who were included in the study.

The authors affirm that human research participants provided informed consent for publication.

## AUTHOR AFFILIATIONS

[1]Department of Pulmonary and Critical Care Medicine, State Key Laboratory of Respiratory Health and Multimorbidity, West China Hospital, Sichuan University, Chengdu, China
[2]Department of Rheumatology and Immunology, Laboratory of Rheumatology and Immunology West China Hospital, Sichuan University, Chengdu, China
[3]Department of Respiratory and Critical Care Medicine, Chengdu First People's Hospital, Chengdu, China
[4]Department of Basic Medicine, Tianfu College, Southwestern University of Finance and Economics, Chengdu, China

## AUTHOR ORCIDs

Lu Cheng http://orcid.org/0000-0001-5875-0339

Zong-an Liang  http://orcid.org/0000-0002-8358-9177
Chun-yu Tan  http://orcid.org/0000-0002-3947-9499

## FUNDING

| Funder | Grant(s) | Author(s) |
| --- | --- | --- |
| Sichuan Association for Science and Technology | 2020YFS0005 | Zong-an Liang |
| International S and T Cooperation Program of Sichuan Province | 2023YFH0081 | Chun-yu Tan |

## AUTHOR CONTRIBUTIONS

Lu Cheng, Validation, Visualization, Writing – original draft, Writing – review and editing | Yan-hong Li, Formal analysis, Writing – original draft | Yin-lan Wu, Data curation, Formal analysis, Methodology | Yu-bin Luo, Conceptualization, Supervision | Yu Zhou, Resources | Tong Ye, Data curation | Xiu-ping Liang, Investigation | Tong Wu, Investigation | De-ying Huang, Investigation | Jing Zhao, Investigation | Yi Liu, Project administration | Zong-an Liang, Conceptualization, Funding acquisition, Project administration | Chun-yu Tan, Conceptualization, Funding acquisition, Supervision

## DATA AVAILABILITY

The raw data supporting the conclusions of this article will be made available by the authors, without undue reservation.

## ETHICS APPROVAL

The study was approved by the ethical committee of West China Hospital of Sichuan University (no. 695 in 2020). Informed consents were obtained from all subjects. All procedures performed in studies involving human participants were in accordance with the ethical standards of the institutional and/or national research committee and with the 1964 Helsinki Declaration and its later amendments or comparable ethical standards.

## ADDITIONAL FILES

The following material is available online.

### Supplemental Material

**Figures S1 to S3 and Tables S1 to S3 (Spectrum00134-25-s0001.docx).** The results of subgroup analysis of IIM patients, disease activity of IIM patients, laboratory tests, and echocardiographic results.

### Open Peer Review

**PEER REVIEW HISTORY (review-history.pdf).** An accounting of the reviewer comments and feedback.

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
