## [Reviewer comments · Microbiology Spectrum]

Microbiology Spectrum

Investigating the Profile of Patients with Idiopathic Inflammatory Myopathies in the Post-COVID-19 Period

Lu Cheng, Yan-hong Li, Yin-lan Wu, Yubin Luo, Yu Zhou, Tong Ye, Xiu-Ping Liang, De-Ying Huang, Tong Wu, Jing Zhao, Yi Liu, Zong-An Liang, and Chunyu Tan

Corresponding Author(s): Zong-An Liang, 四川大学华西医院

Review Timeline:

Submission Date:	January 13, 2025
Editorial Decision:	February 23, 2025
Revision Received:	March 13, 2025
Accepted:	March 25, 2025

Editor: Day-Yu Chao

Reviewer(s): Disclosure of reviewer identity is with reference to reviewer comments included in decision letter(s). The following individuals involved in review of your submission have agreed to reveal their identity: Safdar Ali (Reviewer #1)

Transaction Report:

DOI: <https://doi.org/10.1128/spectrum.00134-25>

Re: Spectrum00134-25 (Investigating the Profile of Patients with Idiopathic Inflammatory Myopathies in the Post-COVID-19 Period)

Dear Dr. Lu Cheng:

Thank you for the privilege of reviewing your work. Below you will find my comments, instructions from the Spectrum editorial office, and the reviewer comments.

Revision Guidelines

Sincerely,
Day-Yu Chao
Editor
Microbiology Spectrum

Reviewer #1 (Comments for the Author):

There are some grammatical mistake highlighted in review, after incorporation of these mistakes, the article will able to publish. All other supplementary and source files are perfect.

Reviewer #2 (Comments for the Author):

It's my pleasure to be invited for reviewing the manuscript. The manuscript compared patients diagnosed as IIM with or without post COVID-19 and found that there was an important association between cardiovascular consequences with IIMs and post COVID-19. Although the result might be correct and believable, there still exist some questions:

- 1.The exact outcomes have not been defined. Including laboratory examination and imageological examination, outcome could be defined as solid description such as death or adverse events.
- 2.More contexts should be added because there are only simple descriptions and not any other further analysis such as subgroups analysis and so on.
- 3.The potential treatment for IIMs with COVID-19 should be mentioned, including cardiovascular drugs like ACEIs and ARBs or rheumatism and immunity drugs.

**Investigating the Profile of Patients with Idiopathic Inflammatory Myopathies in the Post-**

**COVID-19 Period**

Lu Cheng, PhD ^{a,†}, Yan-hong Li, PhD ^{b,†}, Yin-lan Wu, MD ^b, Yu-bin Luo, PhD ^b, Yu Zhou, MD ^c,

Tong Ye, MD ^d, Xiu-ping Liang, PhD ^b, Tong Wu, PhD ^b, De-ying Huang, PhD ^b, Jing Zhao, MD ^b, Yi

Liu, PhD ^b, Zong-an Liang, PhD ^{a,*}, Chun-yu Tan, PhD ^{b,*}

6 ^aDepartment of Pulmonary and Critical Care Medicine, State Key Laboratory of Respiratory Health

and Multimorbidity, West China Hospital, Sichuan University, Chengdu, China.

8 ^bDepartment of Rheumatology and Immunology, Laboratory of Rheumatology and Immunology West

China Hospital, Sichuan University, Chengdu, China.

10 ^cDepartment of Respiratory and Critical Care Medicine, Chengdu First People's Hospital, Chengdu,

China.

12 ^dDepartment of Basic Medicine, Tianfu College, Southwestern University of Finance and Economics,

Chengdu, China.

† These authors contributed equally to this work and shareshared first authorship.

* **Correspondence:** Chun-yu Tan, Department of Rheumatology and Immunology, Laboratory of

Rheumatology and Immunology, West China Hospital, Sichuan University, No.37 Guoxue Alley,

Chengdu 610041, China; And Zong-an Liang, Department of Pulmonary and Critical Care Medicine,

State Key Laboratory of Respiratory Health and Multimorbidity, West China Hospital, Sichuan

University, No.37 Guoxue Alley, Chengdu 610041, China.

**Email:** annaquintessence@163.com (Chun-yu Tan), and liangza@scu.edu.cn (Zong-an Liang)

**Abstract**

**Introduction:** The 2019 coronavirus disease (COVID-19) pandemic has changed the characteristics of
many diseases. It remains unclear whether idiopathic inflammatory myopathies (IIMs) exhibit distinct
phenotypes in the context of COVID-19.

**Methods:** This retrospective study included 171 IIMs patients with a history of COVID-19 (prior
COVID-19, PC) and 121 without (no-prior COVID-19, NPC). Medical histories, lab tests, and
echocardiography data were compared.

**Results:** PC group exhibited a greater incidence of cardiac damage, including a greater proportion of
clinical diagnosis of myocarditis ($p=0.02$), palpitation ($p=0.031$), and MYOACT/MITAX cardiovascular
involvement scores (all $p < 0.001$), and elevated levels of myoglobin (MYO, $p=0.03$), creatinine kinase
MB (CK-MB, $p=0.015$), cardiac troponin T (cTnT, $p=0.011$), N-terminal pro-B-type natriuretic peptide
(NT-proBNP, $p=0.028$), lactate dehydrogenase (LDH, $p=0.033$), and hydroxybutyrate de-hydrogenase
(HBDH, $p=0.019$). Echocardiographic analysis revealed greater diameter of left atrium (LA, $p=0.040$),
left ventricle (LV, $p=0.013$), greater end-diastolic dimension (EDD, $p=0.042$), and greater end-diastolic
volume (EDV, $p=0.036$) in the PC group than in the NPC group. Transcriptional data analysis based on
public databases indicated that various mechanisms, including collagen matrix proliferation, calcium ion
pathway regulation, oxidative stress, cell proliferation, and inflammatory molecules, collectively
contribute to the pathogenesis of myocardial damage in patients with IIMs and COVID-19.

**Conclusion:** The study serves as a crucial reminder for clinicians to remain vigilant regarding the
enduring cardiovascular consequences associated with IIMs ~~subsequent to~~ after COVID-19.

**Keywords:** post-COVID-19, COVID-19, SARS-CoV-2, idiopathic inflammatory myopathies (IIMs),
myocardial damage

***Importance***

[revised manuscript text omitted]

***2.2 Collection of clinical features***

All the data were collected from the electronic medical records database of West China Hospital.
The medical data collected included demographic characteristics, such as age, gender, duration of IIMs,
duration of COVID-19, clinical symptoms and signs, including fever, cough, expectoration, dyspnea,
arthritis, myalgia, myasthenia, and rash, and laboratory features, including routine blood parameters,
erythrocyte sedimentation rate (ESR), C-reactive protein (CRP) level, alanine aminotransferase (ALT)
level, aspartate aminotransferase (AST) level, alkaline phosphatase (ALP) level, gamma
glutamyltransferase glutamyl transferase (GGT), creatine kinase (CK) level, lactate dehydrogenase
(LDH) level, hydroxybutyrate dehydrogenase (HBDH) level, myoglobin protein (MYO) level, cardiac
troponin T (cTnT) level, N-terminal pro-B-type natriuretic peptide (NT-proBNP) level, creatine kinase
isoenzyme MB (CK-MB) level, triglyceride level, cholesterol level, fibrinogen level, antithrombin III
activity, antinuclear antibodies (ANA), anti-Ro52 antibody, myositis specific antibodies, and
echocardiographic parameters.

***2.3 Myositis Disease Activity Assessment***

Disease activity scores were performed by two experienced rheumatologists based on medical
records. Disease activity for IIMs was measured using Myositis Activity Assessment Visual Analog
Scales (MYOACT) and the Myositis Intention-to-treat Activity Index (MITAX), which were established
by the International Myositis Assessment and Clinical Studies (IMACS) group. ~~The detailed~~Detailed
scoring methods can be found in our previous article [8].

**2.4 Analysis of Differentially Expressed Genes**

We retrieved the COVID-19 patient dataset (GEO accession ID: GSE151879), which used the
high-throughput sequencing Illumina NextSeq 500 platform to detect RNA sequences, from which we
sorted the myocardial tissue samples from 3 COVID-19 patients and 3 normal people [9]. The DM patient
dataset (GEO accession ID: GSE143323), including muscle samples from 39 DM patients and 20 healthy
controls, was extracted from RNA sequences using the high-throughput sequencing Illumina HiSeq 3000
~~platform~~platform [10]. Using R software the "DESeq2" and the "edgeR" packages, COVID-19 dataset
(GSE151879) and DM dataset (GSE143323), to $|\text{Log}_2 \text{ Fold Change}| > 0.585$ and $|\text{adj.P.Val. A value of}|$
< 0.05 was used as the implementation standard to find the differentially expressed genes (DEGs) of the
two datasets, and the "Venn" package in R software (4.2.1) was used to identify the common DEGs that
were differentially expressed and upregulated or downregulated in the two datasets at the same time.
Gene Ontology (GO) enrichment analysis and Kyoto Encyclopedia of Genes and Genomes (KEGG)
analysis were performed on DEGs using the enrichment database to reveal the functions of DEGs. The
~~clusterProfiler~~cluster Profiler" package in R software (4.2.1) was used to screen the common DEGs of
potential biological functions and physiological pathways. GO terms included three parts: biological
process (BP), cellular component (CC) and molecular function (MF). A P value < 0.05 and q-value < 0.25
were used as standardized indices.

**2.5 Statistical analysis**

SigmaPlot 12.5 and/or GraphPad Prism 8.0.2 software were used for data analysis. Continuous
variables are presented as the mean \pm standard deviation ($M \pm SD$) or median (Q25, Q75) depending on
whether they fit a normal distribution. Categorical variables are described statistically using percentages.
When the normality (Shapiro–Wilk) test was passed, an independent-samples t test and one-way ANOVA

were conducted for two-group or multiple comparisons, respectively; otherwise, the Mann–Whitney U
test was used for comparison. Chi-square tests were used to compare categorical variables. All tests were
bilateral, and $p < 0.05$ was considered to indicate statistical significance.

**3. Results**

**3.1 Baseline demographics**

The demographics and clinical characteristics of ~~the patients~~patients are presented in Table 1. As
observed, the general baseline characteristics, including age, sex, BMI, and duration of IIMs, were
comparable between two groups (Table 1). The proportions of patients with newly IIMs were 84 (69.42%)
and 109 (63.74%) in the NPC and PC groups, respectively, showing no significant difference ($p > 0.05$).
In the PC group, the time elapsed from prior SARS-CoV-2 infection ranged from 4 weeks to 11 months
(data not shown), with an average duration of 5.92 ± 3.18 months. The proportion of patients with
myocarditis complications in the PC group was significantly greater than that in the NPC group (12.87%
vs. 4.13%, $p = 0.020$). The median time from COVID-19 to the diagnosis of myocarditis was 6 (3,9)
146 months (data not shown in the paper). There were no statistically significant differences in complications
of interstitial lung disease, comorbidity incidence, and treatment between the two groups ($p > 0.050$).

**3.2 Effects of COVID-19 on the clinical characteristics of patients with IIMs**

We analyzed the clinical symptoms of patients with IIMs and observed that the PC group exhibited
a greater prevalence of cardiopulmonary symptoms than did the NPC group (Table 1). These symptoms
included palpitation (21.05% vs. 10.74%, $p = 0.031$) and respiratory rate (20.56 ± 2.10 vs. 20.04 ± 0.98 ,
$p = 0.048$). The NPC group had more rashes (69.42% vs. 52.05%, $p = 0.004$) and arthritis/ arthralgia (40.50%
vs. 26.9%, $p = 0.021$) than did the PC group.

Furthermore, disease activity was assessed in both groups (Figure 2). Compared to those in NPC

group, patients in the PC group had greater MYOACT/MITAX cardiovascular involvement scores (6 (0,
7) vs. 0 (0, 6), $p < 0.001$; 9(0, 9) vs. 0 (0, 9), $p < 0.001$), global MYOACT/MITAX scores (0.37 ± 0.12 vs.
0.35 ± 0.11 , $p=0.046$; $0.29(0.21,0.40)$ vs. $0.27(0.18,0.35)$, $p=0.043$), and MYOACT pulmonary
involvement scores (4.97 ± 3.37 vs. 4.27 ± 3.35 , $p=0.033$). The MYOACT/MITAX cutaneous involvement
scores in the NPC group were greater than those in the PC group (6 (0, 7) vs. 5 (0, 7), $p=0.035$; 3 (0, 3)
vs. 1 (0, 3), $p=0.027$). There was no statistical difference in MYOACT and MITAX scores of other organs
(Supplementary Figure 1).

Subsequently, we ~~analyzed of~~analyzed the laboratory characteristics of the two groups, as presented
in Figure 3 and Supplementary Figure 2. The ratio of anti-aminoacyl transfer RNA synthetase antibody
(ARS) in the PC group was greater than those in the NPC group (24.56% vs. 11.57%, $p=0.009$, Figure
3A). There were no significant differences in the rates of positivity for antimelanoma differentiation
associated gene 5 antibody (MDA5), antinuclear matrix protein 2 antibody (NXP2), anti-helicase protein
antibody (Mi2), anti-small ubiquitin-like modifier activating enzyme (SAE), anti-transcription
intermediary factor 1 γ (TIF1 γ), anti-3-hydroxy-3-methylglutaryl coenzyme A reductase (HMGCR), anti-
signal recognition particle (SRP), or ANA between the two groups (Figure 3A-3B).
[revised manuscript text omitted]

***CRedit authorship contribution statement***

Lu Cheng: Writing – review & editing, Writing – original, draft, Visualization, Validation. Yan-Hong Li:
Writing – original draft, Formal analysis. Yin-Lan Wu: Methodology, Formal analysis, Data curation.
Yu-Bin Luo: Supervision, Conceptualization. Yu Zhou: Resources. Tong Ye: Data curation. Xiu-Ping
Liang: Investigation. Tong Wu: Investigation. De-Ying Huang: Investigation. Jing Zhao: Investigation.
Yi Liu: Project administration. Zong-An Lang: Supervision, Funding acquisition, Conceptualization.
Chun-Yu Tan: Supervision, Funding acquisition, Conceptualization.

***Consent to participate***

Written informed consent was obtained from all individual ~~participants~~participants who were included
in the study.

***Consent for publication***

The authors affirm that human research participants provided informed consent for publication.

***Data availability statement***

The raw data supporting the conclusions of this article will be made available by the authors, without
undue reservation.

***Declaration of ~~competing interest~~Competing Interest***

The authors have no conflicts of interest relevant to this article.

***Acknowledgments***

We would like to thank AJE organization for the English language editing.

***References***

- 1. Davis HE, McCorkell L, Vogel JM, Topol EJ. Long COVID: major findings, mechanisms and
recommendations. Nat Rev Microbiol. 2023 Mar;21(3):133-146. doi: 10.1038/s41579-022-00846-
- 2. Yazdanpanah N, Rezaei N. Autoimmune complications of COVID-19. J Med Virol. 2022
Jan;94(1):54-62. doi: 10.1002/jmv.27292
- 3. Gracia-Ramos AE, Martin-Nares E, Hernández-Molina G. New Onset of Autoimmune Diseases
Following COVID-19 Diagnosis. Cells. 2021 Dec 20;10(12):3592. doi: 10.3390/cells10123592;
- Zacharias H, Dubey S, Koduri G, D'Cruz D. Rheumatological complications of Covid 19.
Autoimmun Rev. 2021 Sep;20(9):102883. doi: 10.1016/j.autrev.2021.102883
- 4. Lundberg IE, Fujimoto M, Vencovsky J, Aggarwal R, Holmqvist M, ChristopherStine L, Mammen
AL, Miller FW. Idiopathic inflammatory myopathies. Nat Rev Dis Primers. 2021 Dec 2;7(1):86.
doi: 10.1038/s41572-021-00321-x.
- 5. Kharouf F, Kenig A, Bohbot E, Rubin L, Peleg H, Shamriz O. Increased rates of idiopathic
inflammatory myopathies during the COVID-19 pandemic: a single-centre experience. Clin Exp
Rheumatol. 2023 Mar;41(2):316-321. doi: 10.55563/clinexprheumatol/970881;

- 6. Apaydin H, Erden A, Güven SC, Armağan B, Karakaş Ö, Özdemir B, Polat B, Eksin MA, Omma
376 A, Kucuksahin O. Clinical course of idiopathic inflammatory myopathies in COVID-19 pandemic:
a single-center experience. *Future Virol.* 2022 May;10.2217/fvl-2021-0146. doi: 10.2217/fvl-2021-
0146
- 7. Lundberg IE, Tjärnlund A, Bottai M, Werth VP, Pilkington C, Visser M, Alfredsson L, Amato AA,
Barohn RJ, Liang MH, Singh JA, Aggarwal R, Arnardottir S, Chinoy H, Cooper RG, Dankó K,
Dimachkie MM, Feldman BM, Torre IG, Gordon P, Hayashi T, Katz JD, Kohsaka H, Lachenbruch
PA, Lang BA, Li Y, Oddis CV, Olesinska M, Reed AM, Rutkowska-Sak L, Sanner H, Selva-
O'Callaghan A, Song YW, Vencovsky J, Ytterberg SR, Miller FW, Rider LG; International Myositis
Classification Criteria Project consortium, The Euromyositis register and The Juvenile
Dermatomyositis Cohort Biomarker Study and Repository (JDRG) (UK and Ireland). 2017
European League Against Rheumatism/American College of Rheumatology classification criteria
for adult and juvenile idiopathic inflammatory myopathies and their major subgroups. *Ann Rheum*
*Dis.* 2017 Dec;76(12):1955-1964. doi: 10.1136/annrheumdis-2017-211468
- 8. Cheng L, Li Y, Luo Y, Zhou Y, Wen J, Wu Y, Liang X, Wu T, Tan C, Liu Y. Decreased Th1 Cells
and Increased Th2 Cells in Peripheral Blood Are Associated with Idiopathic Inflammatory
Myopathies Patients with Interstitial Lung Disease. *Inflammation.* 2023 Feb;46(1):468-479. doi:
10.1007/s10753-022-01747-5
- 9. Yang L, Nilsson-Payant BE, Han Y, Jaffré F, Zhu J, Wang P, Zhang T, Redmond D, Houghton S,
Møller R, Hoagland D, Carrau L, Horiuchi S, Goff M, Lim JK, Bram Y, Richardson C, Chandar V,
Borczuk A, Huang Y, Xiang J, Ho DD, Schwartz RE, tenOever BR, Evans T, Chen S.
Cardiomyocytes recruit monocytes upon SARS-CoV-2 infection by secreting CCL2. *Stem Cell*

- Reports. 2021 Oct 12;16(10):2565. doi: 10.1016/j.stemcr.2021.09.008
- 10. Seto N, Torres-Ruiz JJ, Carmona-Rivera C, Pinal-Fernandez I, Pak K, Purmalek MM, Hosono Y,
Fernandes-Cerqueira C, Gowda P, Arnett N, Gorbach A, Benveniste O, Gómez-Martín D, Selva-
O'Callaghan A, Milisenda JC, Grau-Junyent JM, Christopher-Stine L, Miller FW, Lundberg IE,
Kahlenberg JM, Schiffenbauer AI, Mammen A, Rider LG, Kaplan MJ. Neutrophil dysregulation is
pathogenic in idiopathic inflammatory myopathies. *JCI Insight*. 2020 Feb 13;5(3):e134189. doi:
10.1172/jci.insight.134189
- 11. Xie Y, Al-Aly Z. Risks and burdens of incident diabetes in long COVID: a cohort study. *Lancet*
*Diabetes Endocrinol*. 2022 May;10(5):311-321. doi: 10.1016/S2213-8587(22)00044-4
- 12. Fernández-Ortega MÁ, Ponce-Rosas ER, Muñoz-Salinas DA, Rodríguez-Mendoza O, Nájera
Chávez P, Sánchez-Pozos V, Dávila-Mendoza R, Barrell AE. Cognitive dysfunction, diabetes
mellitus 2 and arterial hypertension: Sequelae up to one year of COVID-19. *Travel Med Infect Dis*.
2023 Mar-Apr;52:102553. doi: 10.1016/j.tmaid.2023.102553
- 13. Raman B, Bluemke DA, Lüscher TF, Neubauer S. Long COVID: post-acute sequelae of COVID-
19 with a cardiovascular focus. *Eur Heart J*. 2022 Mar 14;43(11):1157-1172. doi:
10.1093/eurheartj/ehac031
- 14. Schwartz T, Diederichsen LP, Lundberg IE, Sjaastad I, Sanner H. Cardiac involvement in adult and
juvenile idiopathic inflammatory myopathies. *RMD Open*. 2016 Sep 27;2(2):e000291. doi:
10.1136/rmdopen-2016-00029
- 15. Fairley JL, Wicks I, Peters S, Day J. Defining cardiac involvement in idiopathic inflammatory
myopathies: a systematic review. *Rheumatology (Oxford)*. 2021 Dec 24;61(1):103-120. doi:
10.1093/rheumatology/keab573

- 16. Sherif ZA, Gomez CR, Connors TJ, Henrich TJ, Reeves WB; RECOVER Mechanistic Pathway
Task Force. Pathogenic mechanisms of post-acute sequelae of SARS-CoV-2 infection (PASC).
*Elife*. 2023 Mar 22;12:e86002. doi: 10.7554/eLife.86002
- 17. Artico J, Shiwani H, Moon JC, Gorecka M, McCann GP, Roditi G, Morrow A, Mangion K,
Lukaschuk E, Shanmuganathan M, Miller CA, Chiribiri A, Prasad SK, Adam RD, Singh T,
Bucciarelli-Ducci C, Dawson D, Knight D, Fontana M, Manisty C, Treibel TA, Levelt E, Arnold R,
Macfarlane PW, Young R, McConnachie A, Neubauer S, Piechnik SK, Davies RH, Ferreira VM,
Dweck MR, Berry C; OxAMI (Oxford Acute Myocardial Infarction Study) Investigators; COVID-
HEART Investigators†; Greenwood JP. Myocardial Involvement After Hospitalization for COVID-
19 Complicated by Troponin Elevation: A Prospective, Multicenter, Observational Study.
*Circulation*. 2023 Jan 31;147(5):364-374. doi: 10.1161/CIRCULATIONAHA.122.060632
- 18. Kotecha T, Knight DS, Razvi Y, Kumar K, Vimalasvaran K, Thornton G, Patel R, Chacko L, Brown
JT, Coyle C, Leith D, Shetye A, Ariff B, Bell R, Captur G, Coleman M, Goldring J, Gopalan D,
Heightman M, Hillman T, Howard L, Jacobs M, Jeetley PS, Kanagaratnam P, Kon OM, Lamb LE,
Manisty CH, Mathurdas P, Mayet J, Negus R, Patel N, Pierce I, Russell G, Wolff A, Xue H, Kellman
P, Moon JC, Treibel TA, Cole GD, Fontana M. Patterns of myocardial injury in recovered troponin-
positive COVID-19 patients assessed by cardiovascular magnetic resonance. *Eur Heart J*. 2021 May
14;42(19):1866-1878. doi: 10.1093/eurheartj/ehab075
- 19. Writing Committee; Gluckman TJ, Bhave NM, Allen LA, Chung EH, Spatz ES, Ammirati E,
Baggish AL, Bozkurt B, Cornwell WK 3rd, Harmon KG, Kim JH, Lala A, Levine BD, Martinez
439 MW, Onuma O, Phelan D, Puntmann VO, Rajpal S, Taub PR, Verma AK. 2022 ACC Expert
Consensus Decision Pathway on Cardiovascular Sequelae of COVID-19 in Adults: Myocarditis

and Other Myocardial Involvement, Post-Acute Sequelae of SARS-CoV-2 Infection, and Return to
Play: A Report of the American College of Cardiology Solution Set Oversight Committee. *J Am*
*Coll Cardiol.* 2022 May 3;79(17):1717-1756. doi: 10.1016/j.jacc.2022.02.003

20. Freund O, Eviatar T, Bornstein G. Concurrent myopathy and inflammatory cardiac disease in
COVID-19 patients: a case series and literature review. *Rheumatol Int.* 2022 May;42(5):905-912.
doi: 10.1007/s00296-022-05106-3

21. Shabbir A, Camm CF, Elkington A, Tilling L, Stirrup J, Chan A, Bull S. Myopericarditis and
myositis in a patient with COVID-19: a case report. *Eur Heart J Case Rep.* 2020 Oct 30;4(6):1-6.
doi: 10.1093/ehjcr/ytaa370

22. Duda-Seiman D, Kundnani NR, Dugaci D, Man DE, Velimirovici D, Dragan SR. COVID-19
Related Myocarditis and Myositis in a Patient with Undiagnosed Antisynthetase Syndrome.
*Biomedicines.* 2022 Dec 30;11(1):95. doi: 10.3390/biomedicines11010095

23. Sherif ZA, Gomez CR, Connors TJ, Henrich TJ, Reeves WB; RECOVER Mechanistic Pathway
Task Force. Pathogenic mechanisms of post-acute sequelae of SARS-CoV-2 infection (PASC).
*Elife.* 2023 Mar 22;12:e86002. doi: 10.7554/eLife.86002

24. Artico J, Shiwani H, Moon JC, Gorecka M, McCann GP, Roditi G, Morrow A, Mangion K,
Lukaschuk E, Shanmuganathan M, Miller CA, Chiribiri A, Prasad SK, Adam RD, Singh T,
Bucciarelli-Ducci C, Dawson D, Knight D, Fontana M, Manisty C, Treibel TA, Levelt E, Arnold R,
Macfarlane PW, Young R, McConnachie A, Neubauer S, Piechnik SK, Davies RH, Ferreira VM,
Dweck MR, Berry C; OxAMI (Oxford Acute Myocardial Infarction Study) Investigators; COVID-
HEART Investigators†; Greenwood JP. Myocardial Involvement After Hospitalization for COVID-
19 Complicated by Troponin Elevation: A Prospective, Multicenter, Observational Study.

Circulation. 2023 Jan 31;147(5):364-374. doi: 10.1161/CIRCULATIONAHA.122.060632

[revised manuscript text omitted]

Immunol. 2023 Jan 6;13:1081718. doi: 10.3389/fimmu.2022.1081718

**Legends**

**Table 1. Baseline characteristics of IIMs patients**

*indicates a significant difference between two groups, * $p < 0.05$, ** $p < 0.01$. §The body-mass index (BMI)

is the weight in kilograms divided by the square of the height in meters. #The time from improvement of

COVID-19 related symptoms and negative throat swab results to admission. Abbreviations: COPD,

chronic obstructive pulmonary disease;

**Supplementary Table 1. Electrocardiograph characteristics of patients with IIMs**

# Malignant arrhythmia included ventricular fibrillation, ventricular tachycardia, ventricular arrest,

third degree atrioventricular block, sick sinus syndrome, pre-excitation syndrome with atrial

fibrillation.

**Supplementary Table 2. Characteristics of IIMs patients in different post-COVID periods**

*Indicates statistical difference among the 3 groups, $p < 0.05$. #T means the time from improvement of
COVID-19 related symptoms and negative throat swab results to admission

**Supplementary Table 3. Distribution characteristics of IIMs subtypes.**

*Indicates statistical difference between two groups, $P < 0.05$. Abbreviations: ASS, anti-synthase
syndrome; DM, dermatomyositis; IMNM, immune-mediated necrotizing myopathy; PM, polymyositis;
immune-mediated necrotizing myopathy;

**Figure 1. Flow chart of the clinical study participants.**

IIMs, idiopathic inflammatory myopathies; CTD: connective tissue disease; COVID-19, coronavirus
disease 2019.

**Figure 2. Myositis disease activity score in IIMs patients.**

MYOACT, Myositis Disease Activity Assessment Visual Analog Scales; MITAX, Myositis Intention to
Treat Activity Index. *indicates statistical difference between two groups, $P < 0.05$ ** $P < 0.001$.

**Figure 3. Laboratory findings of IIMs patients on admission to the hospital**

(A)Distribution of myositis specific antibodies (MSA) in IIMs patients. MDA5, antimelanoma
differentiation associated gene 5 antibody; ARS, anti-aminoacyl transfer RNA synthetase antibody;
NXP2, antinuclear matrix protein 2 antibody; Mi2, anti-helicase protein antibody; SAE, anti-small

ubiquitin-like modifier activating enzyme; TIF1 γ , anti-transcription intermediary factor 1 γ ; HMGCR,
anti-3-hydroxy-3-methylglutaryl coenzyme A reductase; SRP, anti-signal recognition particle; (B)
Antinuclear antibody (ANA) expression in IIMs patients. -, ANA negative; +~++, 1: 100 \leq ANA \leq 1:
320; +++, ANA \geq 1: 1000; (C) Anti-cytoplasmic ribonucleoprotein of 52 kDa (Ro52) expression in
IIMs patients; (D)-(G) Markers of myocardial injury expression in IIMs patients; MYO, myoglobin; CK-
MB, creatinine kinase MB; cTnT, cardiac troponin T; NT-proBNP, N-terminal pro-B-type natriuretic
peptide; (H)-(J) Blood biochemical indexes of IIMs patients; LDH, lactate dehydrogenase; HBDH,
hydroxybutyrate dehydrogenase; GGT, gamma glutamyltransferase; (H)-(M) Blood cells from patients
with IIMs; (N) Coagulation index in IIMs patients. FIB, fibrinogen; *indicates statistical difference
between two groups, *P<0.05 **P < 0.01.

**Figure 4. Echocardiographic characteristics of IIMs patients**

LV, left ventricle; LA, left atrium; EDV, end-diastolic volume; EDD, end-diastolic dimension; * indicates
a significant difference between two groups, *P<0.05.

**Figure 5. Bioinformatics analysis of the transcriptomes of muscle tissues from DM patients and** 589 **myocardial tissues from COVID-19 patients.**

(A)Differentially expressed genes in the DM dataset (GSE143323) in the volcanic figure. (B)
Differentially expressed genes in the COVID-19 dataset (GSE151879) in the volcanic figure. (C)Venn
diagram showing the overlap of DEGs between COVID-19 patients and DM patients. (D)Bubble plot of
GO and KEGG functional enrichment analysis results of common DEGs between COVID-19 patients
and DM patients.

**Supplementary Figure 1. Myositis disease activity score in IIMs patients.**

MYOACT, Myositis Disease Activity Assessment Visual Analog Scales; MITAX, Myositis Intention to
Treat Activity Index.

**Supplementary Figure 2. Laboratory findings of patients with IIMs.**

CK, creatine kinase; ALT, alanine transaminase; AST, aspartate aminotransferase; ALP, alkaline
phosphatase; AT-III, antithrombin III; FIB, fibrinogen; CRP, C-reactive protein;

**Supplementary Figure 3. Echocardiographic characteristics of IIMs patients.**

RV, right ventricle; RA, right atrium; MPA, main pulmonary artery; LVPW, left-ventricular posterior
wall; AAO, ascending aorta. IVS, interventricular sep-tum; AV, aortic valve; PV, pulmonary valve; E,
peak velocity of left ventricular early-diastolic fast filling; A, peak velocity of left ventricular late-
diastolic filling; e', velocity of early diastolic myocar-dial movement at mitral ring; a', velocity of late
diastolic myocardial movement at mitral ring; ESD, end-systolic dimension; ESV, end-systolic volume;
SV, stroke volume per minute; EF, ejection fraction; FS, fraction shortening;

**Table 1. Baseline characteristics of IIMs patients**

Characteristic	No prior COVID-19 (NPC, n=121)	Prior COVID-19 (PC, n=171)	p
Female sex, n(%)	95 (79.24)	126(73.68)	0.329
BMI (kg/m²)[§]	22.22(19.83,24.59)	22.48(20.44,25.00)	0.331
Age (years)	51.00(45.40,57.05)	52.00(44.00,58.00)	0.472
First diagnosed IIMs, n(%)	84 (69.42)	109(63.74)	0.377
Disease duration (months)	6.00(3.00,24.00)	10.00(3.00,24.00)	0.489
Time since prior COVID-19 (months)[#]	-	5.92±3.18	-
Comorbidities			
Hypertension, n(%)	11(9.09)	20(11.70)	0.604
Diabetes, n(%)	16(13.22)	21(12.28)	0.952
Coronary artery disease, n(%)	5(4.13)	8(4.68)	0.948
Hyperlipidaemia, n(%)	33(27.27)	37(21.64)	0.331
Cerebral infarction, n(%)	1(0.83)	0	-
COPD, n(%)	2(1.65)	2(1.17)	0.872
Interstitial lung disease, n(%)	70(57.85)	101(59.06)	0.931
Chronic kidney disease, n(%)	1(0.83)	2(1.17)	0.762
Chronic liver disease, n(%)	3(2.48)	9(5.26)	0.378
Anemia, n(%)	24(19.84)	23(13.45)	0.193

Characteristic	No prior COVID-19 (NPC, n=121)	Prior COVID-19 (PC, n=171)	p
Clinical manifestation, n/total (%)			
Fever	23(19.01)	37(21.64)	0.689
Loss of weight	33(27.27)	60(35.09)	0.199
Fatigue	91(75.21)	131(76.61)	0.891
Chest pain	6(4.96)	16(9.36)	0.239
Palpitation	13(10.74)	36(21.05)	0.031*
Shortness of breath/ dyspnea	70(57.85)	115(67.25)	0.129
Rash	84(69.42)	89(52.05)	0.004**
Weakness in the proximal muscles	59(48.76)	80(46.78)	0.830
Myodynia	43(35.54)	68(39.77)	0.541
Arthritis / arthralgia	49(40.50)	46(26.90)	0.021*
Dysphagia	30(24.79)	42(24.56)	0.926
Cough	59(48.76)	77(45.03)	0.610
Expectoration	53(43.80)	69(40.35)	0.639
Heart rate, beats per minute	91.63±14.31	89.42±15.11	0.209
Respiratory rate, breathes per minute	20.04±0.98	20.56±2.10	0.048*
Clinical diagnosis myocarditis, n(%)	5(4.13)	22(12.87)	0.020*
Treatments for IIMs			
Glucocorticoids	50.00(35.00,50.00)	45.00(30.00,50.00)	0.400
Cyclophosphamide	27(22.31)	48(28.07)	0.331
Methotrexate	11(9.09)	18(10.53)	0.837
Mycophenolate mofetil	10(8.26)	14(8.70)	0.931
Ciclosporin A/Tacrolimus	42(34.71)	59(34.50)	0.930
Tofacitinib/Baricitinib	12(9.92)	20(11.70)	0.772
Rituximab	0	3	-
Gamma globulin	11(9.09)	22(12.87)	0.415
Plasmapheresis	0	2	-

Characteristic	No prior COVID-19 (NPC, n=72)	Prior COVID-19 (PC, n=106)	p
Arrhythmia, n(%)	28 (38.89)	39 (36.79)	0.900
Malignant arrhythmia [#] , n (%)	0	3 (2.83)	-
Sinus tachycardia, n (%)	18 (25.00)	22 (20.75)	-
Other types, n (%)	13 (18.06)	17 (16.04)	-

**Supplementary Table 1. Electrocardiograph characteristics of patients with IIMs**

**Supplementary Table 2. Characteristics of IIMs patients in different post-COVID periods**

Characteristic	#T≤3 months(n=51)	3 < T≤6months(n=42)	T > 6months(n=78)	p
MYOACT pulmonary (score)	7.00(6.00, 8.00)	6.00(0, 7.25)	6.00(0,7.00)	0.043*
MYOACT cardiovascular (score)	6.00(3.00, 7.00)	6(0.00, 6.25)	6.00(0, 6.25)	0.395
MITAX pulmonary (score)	9.00(3.00, 9.00)	3.00(0, 9.00)	3.00(0, 9.00)	0.128
MITAX cardiovascular (score)	9.00(0, 9.00)	9.00(0, 9.00)	9.00(0, 9.00)	0.708
MYOACT global (score)	0.42±0.12	0.35±0.11	0.36±0.11	0.004**
MITAX global (score)	0.34±0.14	0.29±0.12	0.29±0.12	0.045*
Thoracalgia, n(%)	6(11.77)	2(4.76)	8(10.26)	0.480
Palpitation, n(%)	10(19.61)	10(23.81)	16(20.51)	0.874
Shortness of breath/ dyspnea, n(%)	39(76.47)	27(64.29)	49(62.82)	0.243
Rash, n(%)	24(47.06)	23(54.76)	42(53.85)	0.693
Arthritis / arthralgia, n(%)	13(25.49)	12(28.57)	21(26.92)	0.946
Heart rate, beats per minute	92.02±16.76	91.05±14.66	86.83±13.94	0.117
MYO (ng/ml)	133.90(23.57, 759.30)	241.50(28.58,610.50)	115.00(30.80,596.50)	0.872
CK-MB (ng/ml)	6.91(1.41,43.05)	12.50(1.21,80.55)	6.22(1.92,75.45)	0.650
cTnT (ng/L)	44.20 (17.70,139.50)	72.55(11.78,246.85)	51.55(16.93,148.50)	0.995
NT-proBNP (ng/L)	148.00(50.00,350.00)	112.00(41.00,301.00)	139.00(75.00,338.25)	0.556
CK (IU/L)	212.00(44.00,1338.00)	281.50(55.05,1525.00)	285.50(54.75,2138.00)	0.847
LDH (IU/L)	397.00(249.00,596.00)	315.00(234.50,75.75)	356.00(278.75,589.75)	0.369
HBDH (IU/L)	302.00(193.00,469.00)	245.00(176.50,402.25)	275.00(210.50,388.25)	0.440
ALP (U/L)	79.00(59.00,97.00)	68.50(55.50,87.00)	73.00(49.75,92.25)	0.271
GGT (U/L)	61.00(19.00, 160.00)	34.00(16.00,76.25)	39.00(20.75,92.00)	0.081
FIB (g/L)	2.58(2.21,3.47)	2.87(2.43,3.42)	2.52(2.15,3.30)	0.254
White blood cell count (×10 ⁹ /L)	7.52(5.06,10.15)	7.95(5.61,10.38)	7.36(5.86,9.44)	0.938
Neutrophils (%)	75.60(68.88,82.00)	70.40(59.90,80.05)	75.60(68.00,82.00)	0.161
Neutrophils count (×10 ⁹ /L)	5.38(3.62,7.61)	5.02(3.55,7.94)	5.30(4.02,6.76)	0.915
Lymphocyte (%)	15.90(9.40,22.20)	18.95(12.08,25.33)	19.00(10.83,23.50)	0.322
Lymphocyte count (×10 ⁹ /L)	1.14(0.77,1.64)	1.30(0.96,2.23)	1.31(0.80,1.89)	0.206
Platelet count (×10 ⁹ /L)	193.65±59.65	211.64±75.86	206.24±79.75	0.443
Anti-Ro52, positive, n/total (%)	24/47(51.06)	20/42(47.62)	36/72(50.00)	0.946
Clinical diagnosis myocarditis, n(%)	9(17.65)	3(7.14)	10(12.82)	0.322
LV, mm	45.67±4.12	46.47±3.52	46.92±3.40	0.209
LA, mm	33.00(29.00,37.00)	32.00(30.00,35.00)	33.00(31.00,36.00)	0.804
RV, mm	20.00(19.00, 22.00)	21.00(20.00,22.00)	21.00(19.00,23.00)	0.126

RA, mm	32.00(30.00,34.00)	32.00(30.00,35.25)	33.50(31.00,37.00)	0.052
IVS, mm	10.00(9.00, 12.00)	9.50(8.00,11.00)	10.00(9.00,11.00)	0.565
LVPW, mm	9.00(8.00, 10.00)	8.50(8.00,10.00)	9.00(8.00,10.00)	0.135
AAO, mm	31.50(28.25,34.00)	32.00(29.00,35.00)	31.00(29.00,34.00)	0.810
MPA, mm	22.00(20.00, 23.00)	21.50(20.00,24.00)	22.00(20.50,24.00)	0.846
E, m/s	0.70(0.60,0.78)	0.70(0.60,0.90)	0.70(0.60,0.90)	0.077
A, m/s	0.80(0.80,1.00)	0.80((0.70,0.90)	0.80(0.60,1.00)	0.992
E/A≤1, n(%)	33(64.71)	22(52.38)	43(55.13)	0.425
AV, m/s	1.30(1.20,1.50)	1.35(1.20,1.70)	1.30(1.20,1.50)	0.712
PV, m/s	1.00(0.80, 1.10)	0.95(0.80,1.10)	0.90(0.80,1.00)	0.580
e', cm/s	6.00(4.00,8.00)	6.00(5.00,8.00)	6.00(5.00,9.00)	0.578
a', cm/s	8.00(7.00, 9.50)	8.50(8.00,10.00)	8.00(7.00,10.00)	0.470
E/ e'	10.00(8.00, 14.00)	12.00(9.00,12.25)	11.00(8.00,14.00)	0.848
E/ e' < 8, n(%)	5(9.80)	3(7.14)	2(2.56)	0.212
EDD, mm	45.00(42.00, 47.75)	47.00(43.75,48.25)	47.00(44.75,49.00)	0.093
ESD, mm	29.00(26.00, 30.75)	28.00(26.00,31.00)	29.00(26.50,32.00)	0.605
EDV, ml	96.75±20.25	99.53±20.20	101.29±16.78	0.464
ESV, ml	31.00(25.00,40.50)	29.00(25.00,39.00)	32.00(26.00,39.00)	0.937
SV, ml	62.07±11.15	67.63±11.50	68.05±11.77	0.021*
EF, %	67.00(61.00,70.50)	70.00(62.75,73.00)	68.00(63.00,73.00)	0.177
FS, %	37.00(33.00,42.00)	39.00(34.00,42.00)	38.00(34.00,42.00)	0.339

**Supplementary Table 3. Distribution characteristics of IIMs subtypes.**

Subtypes of IIMs	No prior COVID-19 (NPC, n=121)	Prior COVID-19 (PC, n=171)	p
ASS	13 (10.74)	37(21.64)	0.023*
DM (non-anti-MDA5 positive)	35(28.93)	32(18.71)	0.057
Anti-MDA5 positive DM	44(36.36)	45(26.32)	0.088
IMNM	21 (17.36)	32(18.71)	0.887
PM	0 (0)	1(0.58)	-
Unclassified IIM	8(6.61)	24(14.04)	0.070

**Figure 1.**

**Figure 2**

Figure 3

Figure 4

Figure 5

**Supplementary Figure 1.**

**Supplementary Figure 2.**

**Supplementary Figure 3.**

It's my pleasure to be invited for reviewing the manuscript. The manuscript compared patients diagnosed as IIM with or without post COVID-19 and found that there was an important association between cardiovascular consequences with IIMs and post COVID-19. Although the result might be correct and believable, there still exist some questions:

1. The exact outcomes have not been defined. Including laboratory examination and imageological examination, outcome could be defined as solid description such as death or adverse events.
2. More contexts should be added because there are only simple descriptions and not any other further analysis such as subgroups analysis and so on.
3. The potential treatment for IIMs with COVID-19 should be mentioned, including cardiovascular drugs like ACEIs and ARBs or rheumatism and immunity drugs.

Dear editor,

We are very glad to receive your email. Thank you and the reviewers for your appreciation and careful reading of the article and providing very detailed and valuable comments to help the revision of the article. We carefully read these comments and revised the manuscript in accordance with them, further clarifying the writing logic to improve the quality of the manuscript. Next, we will respond point by point to the comments of the reviewers and list the details in the manuscript highlighting the changes in the revised manuscript. We sincerely hope to hear from you.

Reviewer #1

Comment : There are some grammatical mistake highlighted in review, after incorporation of these mistakes, the article will able to publish. All other supplementary and source files are perfect.

Response: We sincerely thank you for pointing out the grammatical errors in our manuscript. After receiving your feedback, we have corrected the errors you highlighted. In addition, we have meticulously proofread the entire text, focusing on grammar, spelling, and punctuation to ensure that all spelling and grammatical errors have been rectified. All the revisions are marked in red in our revised manuscript. We apologize for our oversights and will strive to avoid such errors in future submissions.

Reviewer #2

Comment 1: The exact outcomes have not been defined. Including laboratory examination and imageological examination, outcome could be defined as solid description such as death or adverse events.

Response: We sincerely appreciate the valuable comments. Your point about the lack of clear outcome events in the study is critical, and we have carefully rethought the study design and presentation of results. In the study design phase, our main goal was to investigate changes in disease trajectories in IIMs patients during COVID-19. Although we did not specify "outcome events," our study focused on clinical signs and symptoms, IIMs organ involvement and disease activity, laboratory tests, and imaging changes in patients with IIMs. These indicators can be seen to some extent as the "outcome events" of the study. In our study, we found that cardiac damage is an important feature of IIMs patients in long COVID-19. Upon further reflection, we realized the importance of specifying the end event. Therefore, we add the definition of outcome events in the paper: Cardiac injury is diagnosed according to established criteria

derived from prior research, encompassing at least one of the following parameters: (1) elevation of cardiac biomarkers exceeding the 99th percentile upper reference limit. (2) newly detected electrocardiographic abnormalities, including but not limited to arrhythmias (atrial/ventricular tachycardias, atrial/ventricular fibrillations), conduction disturbances, ischemic changes, repolarization abnormalities, or QT interval prolongation. (3) recent echocardiographic findings demonstrating left ventricular systolic dysfunction, impaired ventricular wall motion, pericardial fluid accumulation, or elevated pulmonary arterial pressure. In the RESELTS part, we have reorganized and analyzed the relevant data to present the outcome events more intuitively. Highlighted in red in the revised manuscript in lines 112-118, lines 154-155.

Comment 2: More contexts should be added because there are only simple descriptions and not any other further analysis such as subgroups analysis and so on.

Response: We are deeply grateful for your insightful suggestions, which have significantly enhanced the quality of our study. We have conducted comprehensive subgroup analyses stratified by both the duration of COVID-19 infection and the clinical subtypes of IIMs. These detailed analyses have been systematically documented in the supplementary materials to provide additional insights. While we acknowledge the inherent limitations of this retrospective study, particularly regarding data availability, we have taken proactive measures to address these constraints in our ongoing research. Specifically, in our newly established prospective cohort, we are meticulously documenting COVID-19 infection severity and detailed medication profiles, which we anticipate will yield more robust and clinically meaningful findings in future analyses.

Comment 3: The potential treatment for IIMs with COVID-19 should be mentioned, including cardiovascular drugs like ACEIs and ARBs or rheumatism and immunity drugs.

Response: We sincerely appreciate your valuable feedback. In response to your suggestion, we have conducted a comprehensive review of ACEIs and ARBs administration across all patient cases. The detailed findings have been systematically incorporated into Table 1 to enhance data transparency. Furthermore, we have expanded our discussion in the corresponding section to include a more thorough analysis of cardiac injury management strategies, providing greater clinical context and depth to our therapeutic approach. Highlighted in red in the revised manuscript in lines 264-266, lines 276-285.

Re: Spectrum00134-25R1 (Investigating the Profile of Patients with Idiopathic Inflammatory Myopathies in the Post-COVID-19 Period)

Dear Prof. Zong-An Liang:

The manuscript is accepted but requires minor revisions before publication.

Your manuscript has been accepted, and I am forwarding it to the ASM production staff for publication. Your paper will first be checked to make sure all elements meet the technical requirements. ASM staff will contact you if anything needs to be revised before copyediting and production can begin. Otherwise, you will be notified when your proofs are ready to be viewed.

Sincerely,
Day-Yu Chao
Editor
Microbiology Spectrum

Reviewer #2 (Comments for the Author):

My above problems have been sloved successfully and there are no other problems for me.

I have reviewed the revised version of the following manuscript and the responses have solved my problems in the previous comments. I have no other questions and I think the manuscript have reached the requirement of publishment.